# Abattoir-Based Serological Surveillance and Spatial Risk Analysis of Foot-and-Mouth Disease, Brucellosis, and Q Fever in Lao PDR Large Ruminants

**DOI:** 10.3390/tropicalmed7050078

**Published:** 2022-05-18

**Authors:** Jarunee Siengsanan-Lamont, Watthana Theppangna, Phouvong Phommachanh, Syseng Khounsy, Paul W. Selleck, Nina Matsumoto, Laurence J. Gleeson, Stuart D. Blacksell

**Affiliations:** 1Maihdol-Oxford Tropical Medicine Research Unit, Faculty of Tropical Medicine, Mahidol University, Bangkok 10400, Thailand; jarunee@tropmedres.ac (J.S.-L.); paul.selleck@csiro.au (P.W.S.); laurence.j.gleeson@gmail.com (L.J.G.); 2National Animal Health Laboratory, Department of Livestock and Fisheries, Ministry of Agriculture and Forestry, Vientiane 01001, Laos; wtheppangna@hotmail.com (W.T.); phouvong.phommachanh335@outlook.com (P.P.); 3Department of Livestock and Fisheries, Ministry of Agriculture and Forestry, Vientiane 01001, Laos; s.khounsy@gmail.com; 4Sydney School of Veterinary Science, University of Sydney, Camden, NSW 2570, Australia; nina.matsumoto93@gmail.com; 5Centre for Tropical Medicine & Global Health, Nuffield Department of Medicine, University of Oxford, Oxford OX3 7LG, UK; 6Lao-Oxford-Mahosot Hospital-Wellcome Trust Research Unit, Mahosot Hospital, Vientiane 01000, Laos

**Keywords:** abattoir surveillance, Lao PDR, seroepidemiology, FMD, brucellosis, Q fever, spatial analysis, risk assessment

## Abstract

A national animal disease surveillance network initiated by the Lao PDR government is adopted and reinforced by a joint research project between the National Animal Health Laboratory (NAHL), the Department of Livestock and Fisheries (DLF), and the Mahidol Oxford Tropical Medicine Research Unit (MORU). The network is strengthened by staff training and practical exercises and is utilised to provide zoonotic or high-impact disease information on a national scale. Between January and December 2020, large ruminant samples are collected monthly from 18 abattoirs, one in each province, by provincial and district agriculture and forestry officers. The surveillance network collected a total of 4247 serum samples (1316 buffaloes and 2931 cattle) over this period. Samples are tested for antibodies against *Brucella* spp., *Coxiella burnetii* (Q fever) and Foot-and-Mouth Disease Non-Structural Protein (FMD NSP) using commercial ELISA kits and the Rose Bengal test. Seroprevalences of Q fever and brucellosis in large ruminants are low at 1.7% (95% CI: 1.3, 2.1) and 0.7% (95% CI: 0.5, 1.0) respectively, while for FMD NSP it is 50.5% (95% CI: 49.0, 52.0). Univariate analyses show differences in seroprevalences of Q fever between destination (abattoir) province (*p*-value = 0.005), province of origin (*p*-value = 0.005), animal type (buffalo or cattle) (*p*-value = 0.0008), and collection month (*p*-value = 3.4 × 10^−6^). Similar to Q fever, seroprevalences of brucellosis were significantly different for destination province (*p*-value < 0.00001), province of origin (*p*-value < 0.00001), animal type (*p*-value = 9.9 × 10^−5^) and collection month (*p*-value < 0.00001), plus body condition score (*p*-value = 0.003), and age (*p*-value = 0.007). Additionally, risk factors of the FMD NSP dataset include the destination province (*p*-value < 0.00001), province of origin (*p*-value < 0.00001), sex (*p*-value = 7.97 × 10^−8^), age (*p*-value = 0.009), collection date (*p*-value < 0.00001), and collection month (*p*-value < 0.00001). Spatial analyses revealed that there is no spatial correlation of FMD NSP seropositive animals. High-risk areas for Q fever and brucellosis are identified by spatial analyses. Further investigation of the higher risk areas would provide a better epidemiological understanding of both diseases in Lao PDR. In conclusion, the abattoir serological survey provides useful information about disease exposure and potential risk factors. The network is a good base for field and laboratory staff training in practical technical skills. However, the sustainability of such a surveillance activity is relatively low without an external source of funding, given the operational costs and insufficient government budget. The cost-effectiveness of the abattoir survey could be increased by targeting hotspot areas, reducing fixed costs, and extending the focus to cover more diseases.

## 1. Introduction

High-impact animal diseases and zoonoses remain a threat to food security and public health, especially in low- and medium-income countries where a large proportion of the population relies on agricultural production for their livelihoods [1]. Over the past decade, many outbreaks of such infectious diseases, including emerging, newly introduced, and endemic diseases, have been reported in Southeast Asia (SEA). Transboundary animal diseases have been estimated to cause the region billions of dollars in losses yearly [2]. In Lao PDR, buffaloes and cattle are widely raised by smallholders to provide additional sources of income [3,4]. The Lao PDR government has declared that one of the country’s priorities is to eradicate poverty, with the national agricultural development strategy aiming to achieve food security through strengthening agricultural production [5]. The capabilities of field and laboratory personnel in disease detection, monitoring, diagnosis, and biosafety and biosecurity are critical for disease control and prevention. Strengthening capacity in these areas will boost livestock productivity and smallholder income and contribute to national food security.

Neglected zoonoses, including brucellosis and Q fever (caused by *Coxiella burnetii*), have been previously reported in livestock in Lao PDR [6]. However, information on the disease circulation in Lao PDR’s livestock populations and potential impacts on humans are still widely unknown. In ruminants, Q fever often causes mild disease, but in some cases, abortion and stillbirths can occur [7]. *C. burnetii* can be transmitted to humans by inhalation of contaminated dust or by drinking unpasteurised milk [7]. Brucellosis causes productivity losses, including abortion, infertility, milk reduction, and unhealthy calves [8] and can be transmitted to humans by direct contact via mucous membranes or open wounds and ingestions of infected products [9]. In humans, both agents can cause non-specific symptoms, including undulant fever, while Q fever can also cause chronic diseases, which are difficult to diagnose [10]. A study in northeastern Thailand estimated a Q fever prevalence of 1.3% in patients presenting to four hospitals with fever [11]. Another study reported that human brucellosis cases were reported almost every year in Thailand, and 34 livestock officers in a southern province showed a *Brucella* seroprevalence of 8.8% [12]. Awareness of brucellosis and disease transmission among small-scale farmers was reported to be low [13,14]. Thus, these pathogens pose a health risk to people who have close contact with livestock.

Foot-and-Mouth Disease (FMD) is a high-impact animal disease-causing significant productivity and economic losses [15] and limiting international trade. Lao PDR is a partner country in the Southeast Asian and China FMD control programme (SEACFMD) coordinated by the World Organisation for Animal Health (OIE), and there is a strong interest within the SEACFMD to strengthen FMD surveillance and improve epidemiological understanding of the disease [16]. In addition, there is strong regional demand for beef for human consumption, and Lao PDR would like to participate in the trade, but the lack of information about FMD is a constraint on participation. Animal disease surveillance programmes in Lao PDR are mostly funded by international agencies and donors. The last nationwide animal disease survey was conducted in 1999–2001 with the main focus on FMD [15]. Since then, animal disease survey programs in Lao PDR have been conducted targeting specific provinces and areas of interest.

In 2018, the establishment of the national animal disease surveillance network was initiated by the Lao PDR government and adopted by a joint project between the National Animal Health Laboratory (NAHL), the Department of Livestock and Fisheries (DLF), and the Mahidol Oxford Tropical Medicine Research Unit (MORU) to enhance animal disease detection and biosafety and biosecurity capabilities. The network was utilised by our study as a part of capacity building activities for government personnel to practise collecting animal samples and biodata, processing and submitting the samples to the NAHL, applying biosafety and biosecurity principles in fieldwork, and performing laboratory diagnoses of surveillance samples. The outcomes of the nationwide surveillance could also be used to provide an up-to-date status for several diseases in Lao PDR. This network focused on brucellosis, Q fever, and FMD. Both zoonoses had been reported in Lao PDR livestock; however, more information on the disease distributions and risk factors would benefit future One Health research and raise awareness. The information on FMD distribution will contribute to the country’s FMD control program. This study used the surveillance data to analyse the disease status and risk factors, identify spatial hotspots and gaps in the national surveillance network, and provide recommendations for future sustainability.

## 2. Materials and Methods

### 2.1. Field Sample Collection

The surveillance was conducted between January and December 2020 in all eighteen provinces (one prefecture and 17 provinces) throughout Lao PDR by provincial teams comprised of trained District Agriculture and Forestry Office (DAFO) and Provincial Agriculture and Forestry Office (PAFO) personnel. Blood samples were collected from an abattoir in each province once a month between January and May as a trial period and twice a month from June to December to maximise information. The sample size calculation of eleven animals was previously described by Siengsanan-Lamont et al. [6] using an estimated abattoir herd size of 30, an estimated prevalence of 20%, a diagnostic test sensitivity of 99%, and a confidence level of 95%. As abattoir herd sizes vary between provinces, the teams were directed to collect 10 mL of blood from up to 20 large ruminants per collection round. No preference was given regarding species of the ruminants as that was subject to availability on the day. Blood samples were transported back to the provincial offices in a cooler with ice or ice packs (at approximately 4 °C). Serum samples were prepared without a centrifuge by the provincial teams using a technique of standing the blood sample for 2–12 h at room temperature or in a 4 °C refrigerator overnight and then decanting the serum. Serum samples were placed in labelled cryotubes and shipped to NAHL in a cooler. Once received at NAHL, the samples were checked and stored in a −30 °C freezer. Records of the samples were entered into the NAHL database.

### 2.2. Serological Diagnostic Testing

Commercial enzyme-linked immunosorbent assay (ELISA) kits from IDvet were used to detect antibodies against *Brucella* spp. (ID Screen^®^ Brucellosis Serum Indirect multi-species, Cat# BRUS-MS-10P), FMD Non-Structural Protein (NSP) (IDScreen^®^ FMD NSP Competition, Cat# FMDNSPC-10P) and *Coxiella burnetii* (ID Screen^®^ Q Fever indirect multi-species, Cat# FQS-MS-5P). ELISAs were performed by trained NAHL staff according to the manufacturer’s protocols provided with the kits. IDSoft^TM^ software (version 5.05; [17]) provided with the ID Screen^®^ ELISA kits was used to calculate the Sample to Positive Ratio (S/P%) and competition percentage (S/N%). The S/P% and S/N% cut-off points and diagnostic test sensitivity and specificity of the commercial kits were detailed in the manufacturer’s protocols and were as previously applied by Siengsanan-Lamont et al. [6,18]. The Rose Bengal test (RBT) was used to confirm the positive *Brucella* antibodies ELISA samples [19]. However, some sera were tested only by RBT as the ELISA kit was in short supply due to COVID-19-related supply chain issues.

### 2.3. Statistical, Spatial Correlation, and Relative Risk Analyses

Data analyses were conducted using Microsoft Excel and RStudio Version 1.2.1335 [20]. Descriptive statistics, including frequency and probability distributions, were used to describe the dataset. The leaflet R package [21] was used to generate visualisations of animal origins. The epiR package [22] was used for the calculation of apparent and true seroprevalences using the Wilson method for imperfect tests [23]. Chi-square and multivariate logistic regressions were used for risk factor analyses and odds ratio (OR), including 95% confidence intervals (CI) [24]. The independent variables included destination (abattoir) province, province of origin, sex, body condition score (BCS), age, animal type (buffalo or cattle), collection date, and collection month. The independent variable was fitted in a univariate analysis against each test result. Variables with a *p*-value < 0.1 were included in the multivariate analyses. The final model was selected using the *p*-value of variables (≤0.05), Akaike information criterion, Hosmer–Lemeshow goodness of fit test and variance inflation factor (VIF) testing multicollinearity.

Livestock census 2011 data (available from http://www.decide.la/, accessed on 18 August 2021) and a constant cattle growth rate of 5% per year and buffalo growth rate of −2.0% per year [4] were used to calculate an estimate of the total number of cattle and buffalo for 2020. The standardised morbidity rate (SMR) of each province was then calculated using estimated figures and the following formula. Xaisomboun was omitted from the spatial analyses.
$$\mathrm{SMR}=\frac{Observed\;disease\;animals}{Expected\;disease\;animals}=\frac{estimated\;population\;per\;province\times FMD\;seroprevalence\;of\;the\;province}{estimated\;population\;per\;province\times the\;average\;FMD\;seroprevalence}$$

Spatial disease relative risks were identified using the Besag–York–Mollié (BYM) model [25] in the INLA R package [26]. The SMR maps were plotted compared to the mean posterior relative risk (MPRR) maps.

## 3. Results

During the initial pilot period from January to May, samples were only collected once a month as a network trial period, and then the frequency was increased to twice a month as the network’s proficiency increased. The field and laboratory staff who participated in the surveillance network demonstrated competency in performing their assigned tasks of sample collection, preparation, submission, and laboratory testing. A total of 4247 samples were collected from 1316 buffaloes and 2931 cattle during the surveillance program. The total samples collected by each province varied according to the availability of animals at the abattoir on the collection days (Figure 1). In the Xaisomboun province, participation was limited by a number of local factors, with the team only participating between January and May 2020. The remaining provincial teams collected samples throughout the year, except in April 2020, when the Lao PDR government imposed a countrywide COVID-19 lockdown. The average cost per sample of field consumables and their distribution was USD 1.4, while the total cost of fieldwork (combining staff per diem, logistics of consumables and samples, and field consumables) was approximately USD 3.8. The majority of buffalo samples (*n* = 1316) were from animal age groups 5–6 years old (28.9%), 7–8 years old (24.2%), and 9 years old or more (25.1%). For cattle (*n* = 2931), approximately 29.9% of the samples were from the age group between 3–4 years old, 33.8% between 5–6 years old, and 14.0% between 7–8 years old (Figure 2). The highest numbers of buffalo (163 or 12.4%) and cattle (351 or 11.9%) samples were collected in June and September, respectively (Figure 3). Seroprevalences in both Figure 2 and Figure 3 were plotted in negative log scale. Phongsali province supplied the highest number of buffalo samples (182 or 13.8%), while Savannakhet province supplied the most cattle samples (434 or 14.9%). Even though Vientiane prefecture collected the highest number of buffalo samples, the prefecture supplied no buffaloes.

Overall, the apparent and true seroprevalences are present in Table 1. The seroprevalences of antibodies against the three pathogens per age group and per collection month are shown in Figure 2 and Figure 3. The prevalence of FMD NSP, Q fever, and *Brucella* antibodies in buffaloes (*n* = 1316) was 48.8%, 0.8%, and 0.1%, and in cattle (*n* = 2931) were 51.2%, 2.1%, and 1.0%, respectively. The highest FMD NSP seroprevalences were found for samples collected in November (61.0%, *n* = 503), September (61.0%, *n* = 479), and October (55.8%, *n* = 477), while March showed the lowest seroprevalence (40.5%, *n* = 237). The Q fever seropositive samples were detected throughout the year but peaked in May (4.3%, *n* = 232). The greatest number of the Q fever seropositive cattle (11 out of 71 positives) originated from Luang Prabang province, of which 10 out of 11 were collected in August. Out of 30 seropositive brucellosis samples, 28 samples were collected in September, and almost all of them (27) originated from Luang Prabang province (Table 2). Based on the BCSs scaling from one to five, seropositive brucellosis animals had a BCS of three (20 out of 30) and 4. The detail of seroprevalence by provinces of origin are presented in Table 3. The seroprevalence of FMD NSP antibodies varied by animal type and province of origin. The overall FMD NSP seroprevalence in cattle (51.2%, *n* = 2931) was higher than in buffaloes (48.8%, *n* = 1316). The provinces of origin with the highest FMD NSP seroprevalence were Khammouane (79.0%, *n* = 295), Bokeo (73.6%, *n* = 110), and Houaphanh (62.1%, *n* = 322). Visualisations of the large ruminant movement from the point of origin (black dot) to the destination (abattoir) province (grey dot) were presented in Figure 4. The Xaisomboun province was excluded from the visualisation (Figure 4) as the total number of animals was low (*n* = 5) and locally sourced. Only eight of the Lao provinces, including Xaisomboun, sourced ruminants within their province and immediate neighbouring provinces. The remainder received some animals from long-distance sources. Moreover, only one buffalo originated from Thailand.

Univariate logistic regression analysis of the FMD NSP dataset showed that significant variables for seropositive status included the destination province, province of origin, sex, age, collection date, and collection month (Table 4). Large ruminants originating from Khammouane were 8.4 (OR, 95% CI: 5.9, 12.1) times more likely to be FMD NSP seropositive animals compared to those originating from Phongsali (reference group). Animals aged 9 years old or more had a higher chance of having FMD NSP antibodies (OR 1.5, 95% CI: 1.3, 1.6) than those aged 1–2 years old. The number of FMD NSP positive samples collected in September and November was significantly higher (OR 2.3, 95% CI: 1.7, 3.2) than those collected in March. For the brucellosis and Q fever datasets, cattle were more likely to have antibodies to *Brucella* spp. (OR 13.1, 95% CI: 1.8, 96.5), and *C. burnetii* (OR 2.8, 95% CI: 1.4, 5.4) than buffaloes (Table 4). There were significant differences in seroprevalences of Q fever between destination province, province of origin, and collection month, while for *Brucella* between destination province, province of origin, BCS, age, and collection month.

The 2011 census reported a total of 1.58 million cattle and 774,200 buffaloes in Lao PDR. Thus, the estimated cattle and buffalo populations in 2020 were 2.45 million and 645,490, respectively. Figure 5 shows the SMR (left) and the MPRR (right) plots of brucellosis, Q fever, and FMD NSP. For brucellosis, Luang Prabang province had the highest SMR; however, the high MPRR areas were Luang Prabang (MPRR = 18.67, 95% CI: 10.42, 40.99), Xiangkhouang (MPRR = 15.52, 95% CI: 7.28, 37.85), Phongsali (MPRR = 15.35, 95% CI: 7.28, 37.85) and Salavan (MPRR = 15.32, 95% CI: 7.07, 37.64) provinces. For Q fever, high SMR and MPRR provinces identified by the spatial modelling included the following: Oudomxay (MPRR = 2.45, 95% CI: 0.70, 4.33), Luang Namtha (MPRR = 2.45, 95% CI: 0.71, 4.33), Luang Prabang (MPRR = 2.22, 95% CI: 0.48, 4.10), Xiangkhouang (MPRR = 2.08, 95% CI: 0.34, 3.96), Salavan (MPRR = 1.88, 95% CI: 0.14, 3.76), Xekong (MPRR = 1.75, 95% CI: 0.01, 3.63), and Houaphanh (MPRR = 1.74, 95% CI: 0.00, 3.62). For FMD NSP, Bokeo, Viantiene prefecture, and Khammouane provinces had the highest SMR and were also higher MPRR areas compared to other provinces. However, the FMD NSP MPRR of provinces ranged between −0.49 and 0.69.

## 4. Discussion

NAHL and some provincial offices had staff-workload imbalance issues. The NAHL laboratory staff were not only responsible for the government’s routine diagnostic work but also provided diagnostic services to all animal health externally funded projects, which had overwhelmed the laboratory management system, especially the inadequate specimen inventory system. Moreover, consumable stocks at provincial offices were often inadequate, e.g., personal protective equipment (PPE), sample collection tubes, etc. Some of the consumables supplied for the abattoir surveillance were diverted for outbreak investigations. Due to the absence of established in-country suppliers, consumables used in the survey were shipped from Thailand. The distribution of consumables shipped from NAHL in Vientiane to the provinces, especially the remote areas, was costly. Moreover, the total samples collected by the network in 2020 were 20% less than the calculation in the study design. Thus, the average cost of field consumables per sample was higher compared to those reported in our previous study [6]. Furthermore, many animal health-related activities funded by international agencies were required to pay government staff a standardised per diem for their participation. The fixed labour cost per sample collection per day was relatively high (~USD 26/person), given that the minimum monthly salary for government officers was USD 132 [27]. This requirement could negatively affect the long-term sustainability of the surveillance network as staff may expect to be financially supplemented to perform their routine responsibilities, e.g., disease monitoring and reporting. This distortion of work practices is one of the occasional criticisms of externally supported projects in the resource-poor animal health sector in some countries in Southeast Asia.

In our study, despite the calculated sample size of 11, officers were directed to collect up to 20 samples per collection round. It was due to abattoirs in Lao PDR’s major urban areas were reported to slaughter between 8–60 cattle and buffaloes per day [28]. However, in smaller provinces, the numbers of buffaloes and cattle slaughtered at abattoirs were expected to be much smaller than the above figure. The sample size of up to 20 animals was used on the basis that a minimum of 30% or more of the total animals presented at the abattoir will be sampled. The number and type of animals collected per province varied depending on the availability of abattoir animals on the collection days. Provinces with a high human population, such as Luang Prabang, Champasak, Savannakhet, Vientiane, and Vientiane prefecture [29], were expected to collect more samples than smaller provinces where low numbers of animals were slaughtered at the local abattoir per day. However, Vientiane province collected fewer samples than expected, while Phongsali collected more samples, indicating possible favourable factors such as staff enthusiasm and/or availability. As the probability sampling method was not applied, the outcomes of our study did not extrapolate to the entire population. Unlike Cambodia, where buffaloes were rarely slaughtered at abattoirs [30], in Lao PDR, buffaloes were regularly slaughtered at abattoirs throughout the country. Based on the FAOSTAT website, in 2019, Lao PDR produced four times more buffalo meat than Cambodia [31], indicating a higher demand for buffalo meat.

The overall seroprevalence of Q fever in 2020 was lower than that of the six-province survey in 2019 at 2.2% [6] but higher than the seroprevalence of a survey carried out in seven provinces reported in 2016 at 1.2% [32]. *C. burnetii* infection in humans commonly causes acute fever, pneumonia and hepatitis, and endocarditis in chronic infection [11]. Some publications reported investigations of *C. burnetii* in Lao PDR patients with non-malarial febrile illness or possible endocarditis; however, the agent was not detected [33,34]. In neighbouring Thailand, Q fever has been reported as widespread since 1966 [11]. A survey of patients admitted to hospitals in northern and northeastern Thailand between 2002–2005 reported a seroprevalence of 0.5% [35]. Even though Q fever seroprevalence in large ruminants in Lao PDR was low, the infection was geographically widely distributed in large ruminants. Moreover, Q fever seroprevalence was higher in the Lao PDR goat population at 4.1% [10]. The extent to which the Q fever infection in cattle is dependent on the presence of goats has not been investigated. Other domestic animals, e.g., dogs, cats, and pigs, are also known reservoirs of the agent [36] and must also be considered in an outbreak situation. Thus, the impacts of the disease on human health in Lao PDR are currently unknown.

Similar to the 2006 [37], 2013–2015 [32], and 2019 [6] surveys, the prevalence of *Brucella* reactors in large ruminants was low and area-specific. However, Olmo [38] reported that no *Brucella* sero-reactors were detected from 61 buffalo and 90 cattle survey samples collected from six provinces after 2012 and stored at NAHL. Most *Brucella* seropositive animals in our study originated from Luang Prabang and were collected in September. Seroprevalence of brucellosis in goats in Lao PDR in 2015–2016 was 1.4% [10], which was higher than our study in large ruminants. In contrast, the prevalence of *Brucella* spp. reported in Chiang Rai, Thailand was 3.3% in dairy cattle [39]. Other studies in Thailand reported that seroprevalences were 1.4% in goats and 1.6% in sheep in 2013 [40] and overall 0.72% in small ruminants between 2013 and 2015 [41]. It is interesting that brucellosis is not a prevalent infection in cattle in this tropical, low-density production system. Moreover, data on *Brucella* spp. in humans in Lao PDR was not available [38]. Further investigation of the sources of the positive cluster would help better understand local epidemiology as vaccination was not available.

The *Brucella* and Q fever seropositive animals in our study were aged more than one year and older. Both diseases affect reproductive animals, and large amounts of bacteria can be found in birthing products (e.g., placenta, aborted fetus, and vaginal fluid, etc.) [8,9]. Other animals can be infected by the ingestion (and inhalation for Q fever) of contaminants. A study reported that Q fever was rather a horizontal transmission, and the prevalence of the disease was significantly higher in older animals [42]. Lao PDR has the following two main seasons: wet (May–October) and dry (November–April). Cattle and buffaloes breeding cycles have been described, usually with mating in the late dry–early wet seasons and calving in the early dry season [28]. The differences in disease seroprevalence between collection months may be influenced by these mating and calving seasons. In our study, *Brucella* seropositive animals were sampled between May and September, while the Q fever seroprevalence in the abattoir animals also peaked in June and August, considered the mating season.

The overall FMD NSP seroprevalence in large ruminants detected in our study (54.6%) was higher than those of the abattoir surveillance conducted in 2019 (41.6%, *n* = 683 [18]), the survey in nine northern provinces in March 2019 (43.7%, *n* = 602 [43]) and another survey in three provinces in 2018 (43.0%, *n* = 684 [44]). The FMD NSP ELISA test only detects antibodies produced by natural infection or a less likely case of repeated vaccination, or if animals have received an unpurified vaccine [45]. However, a natural infection could result in FMD NSP antibodies that persisted for more than five years [46], and so the FMD NSP seroprevalence likely represented the cumulative prevalence of the exposure. Moreover, FMD NSP seropositive animals originated from all provinces, indicating that FMD viruses were widely distributed in Lao PDR. Similar to our study, albeit with a much smaller sample, in the 2019 study, the animals that originated from Khammouane had the highest FMD NSP seroprevalence (70.0%, *n* = 10) [18]. Our study did not include FMD vaccination history, which may have impacted seroprevalences in the origin provinces. In this type of surveillance, vaccination history is not likely to be available, as traders do not concern themselves with this information, and it is not required on movement permits. Interestingly, the FMD NSP seroprevalence of the 4–6-month-old group was the highest in our study. Even though older animals were more likely to be exposed to FMD viruses or be vaccinated multiple times in their lifetime, calves up to six months old could also have maternal immunity from vaccinated or infected dams [45]. The FMD vaccination campaigns funded by the Lao PDR government and international organisations were implemented in some selected provinces between 2012 and 2016 [47] and 2018 [44]. In the areas outside the vaccination campaigns, farmers can also vaccinate their animals at their own cost; however, not many farmers do this, and a vaccine is not readily available [44]. Thus, FMD vaccination coverage in Lao PDR is limited [48], and generally, animals do not receive booster vaccinations every six months, resulting in inadequate immune protection [44]. The question thus arises as to the best strategy for FMD control in Lao PDR. Vaccination programmes are very expensive, but if there was an economic incentive to control the disease, then local authorities might be more engaged and even champion the programme. The FMD serosurveillance reported here would suggest that there is a low-level endemic circulation of FMD viruses in Lao PDR, and outbreak reports possibly depend on the occurrence of disease above a tolerated threshold. The dynamics of endemic disease and the cycle of herd immunity might be the determinants of outbreak recognition and reporting. The incursion of a new strain of the virus can also upset the balance and trigger a significant outbreak. A further investigation of FMD prevalence and vaccination history that targeted the high seroprevalence provinces of origin would facilitate a clearer understanding of FMD virus epidemiology in Lao PDR.

No samples from cattle originating from Thailand were collected in our study, which was similar to our previous study [6] and indicated that reported livestock transboundary movements from Thailand to Vietnam and China via Lao PDR [49,50,51] are not used to supplement the local trade. The movement visualisations showed that some provinces, including Bolikamxai, Luang Namtha, Xiangkhouang, and Vientiane prefecture, sourced animals from longer distances compared to other provinces. This indicated a high demand for meat, especially in the highly populated Vientiane prefecture. However, for the provinces with smaller populations, including Luang Namtha and Xiangkhouang, this perhaps indicated replacements for cross-border trade activities. Presently, there is no market incentive to develop a better system to track livestock (e.g., tags, microchips, etc.) that would provide better information about the animal’s origin and vaccination history and could contribute to more effective national and regional control programmes for transboundary diseases. In overall terms, the challenges faced in improving disease surveillance in large ruminants in resource-limited settings are considerable and remain a constraint to regional transboundary disease control.

The provinces of origin and destination were identified as FMD risk factors, in particular Khammouane and Bokeo, where OR was high. Both provinces were identified as one of the transit routes of livestock trade from Thailand to Vietnam and/or China [51]. Even though our abattoir ruminants were recorded as local livestock, if transited animals were actively infected, they may have spread the disease to local stocks. Risk factors and odds ratios identified in our study could provide insights into the variables that were influencing the seroprevalences of the diseases. However, these risk analyses had limitations as there were other factors outside of our study scope that were not included in the analyses. Tenny and Hoffman [52] suggested that the odds ratios would overestimate the risk in the case of common diseases. Thus, for the FMD data, relative risks may be more accurate than odds ratios. However, the calculation of relative risk was not possible with the type of passive surveillance conducted.

The BYM model used in our study was to test the spatial correlation between positive animals in neighbouring provinces and other areas [25]. Based on the MPRR maps, the highest MPRR of brucellosis was as high as 18 compared to the highest MPRR of Q fever at 2. This was due to brucellosis being relatively rare and only detected in animals originating from four provinces compared to positive Q fever animals detected in 15 provinces of origin. However, it is unclear why Q fever high-risk areas were in the northern and southern provinces only. Thus, we recommended that future investigation of brucellosis and Q fever should target these high-risk areas. When comparing our maps (Figure 5) with previously published livestock movement maps [49,51,53], the FMD NSP SMR and disease risk maps were in line with the trade movements. Even though some provinces showed higher MPRR for FMD NSP, none of these MPRRs was higher than one, meaning no spatial difference of seropositivities between provinces. This indicated that animals with FMD NSP antibodies were relatively uniformly distributed throughout Lao PDR.

## 5. Conclusions

The abattoir serological survey provided useful information about disease exposures and identified risk factors. The network provided useful training exercises for field and laboratory staff to practise technical skills. However, the sustainability of such a surveillance activity was relatively low without an external source of funding, given the high operation costs and insufficient government budget. The cost-effectiveness of the abattoir survey could be increased by targeting hotspot areas, reducing fixed costs, including staff per diem, and extending the focus to cover more diseases. The seroprevalence of brucellosis and Q fever in cattle and buffalo were relatively low and areas specific. While for FMD NSP, the seroprevalence was high and consistent across the country. Given that the diseases present in livestock and the impacts of brucellosis and Q fever on public health in Lao PDR are unknown, multidisciplinary research should be conducted using the One Health approach. Further investigation of the high-risk areas where positive animals originated would provide a better understanding of the epidemiology of the diseases.

## Figures and Tables

**Figure 1 tropicalmed-07-00078-f001:**
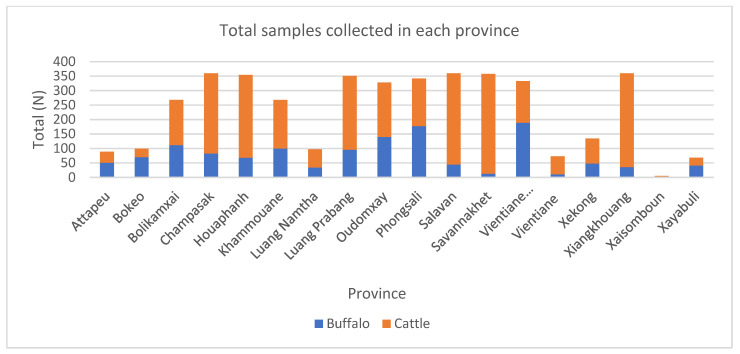
Total samples collected in each destination province.

**Figure 2 tropicalmed-07-00078-f002:**
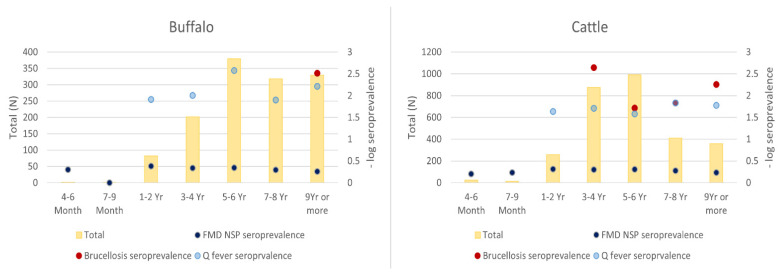
Total numbers of animals per age group versus seroprevalence (−log scale) rate.

**Figure 3 tropicalmed-07-00078-f003:**
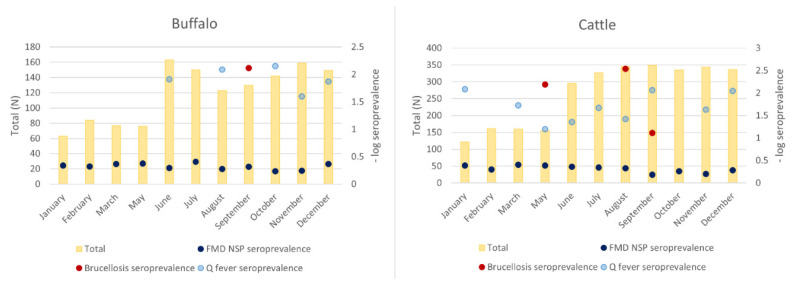
Total numbers of animals per collection month versus seroprevalence (−log scale) rate.

**Figure 4 tropicalmed-07-00078-f004:**
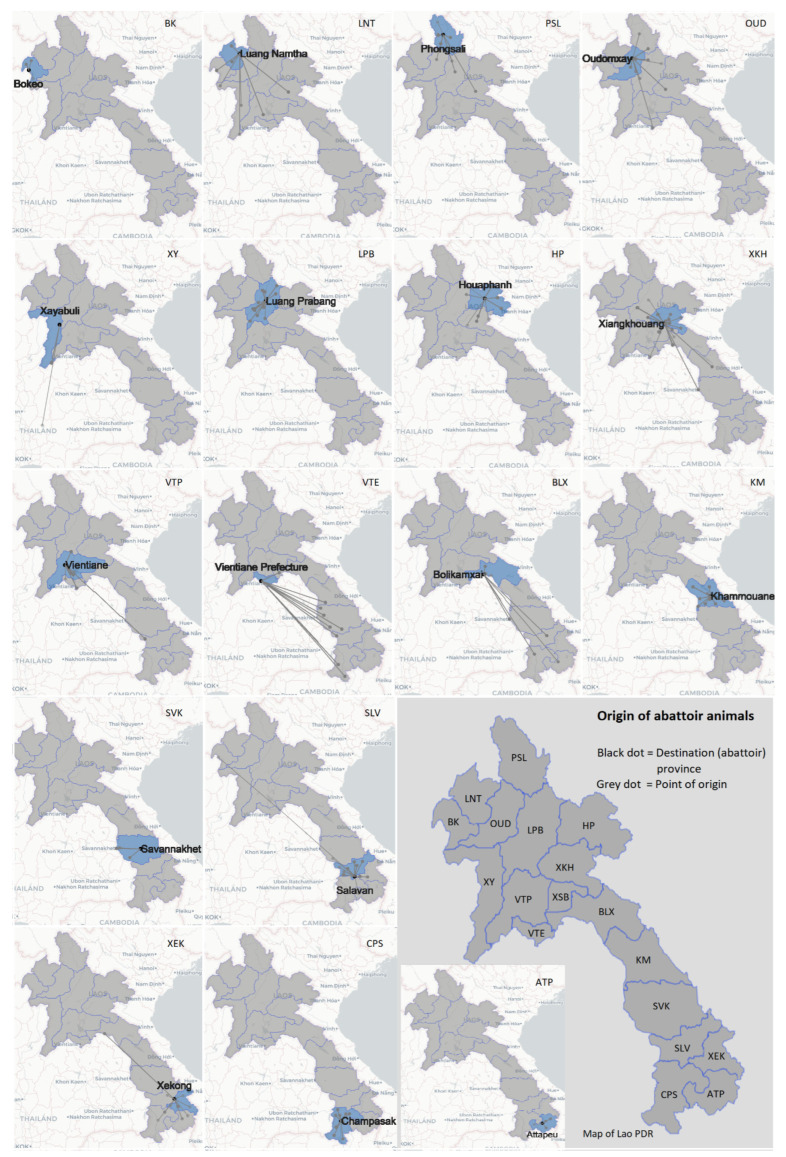
Visualisation of animal origins.

**Figure 5 tropicalmed-07-00078-f005:**
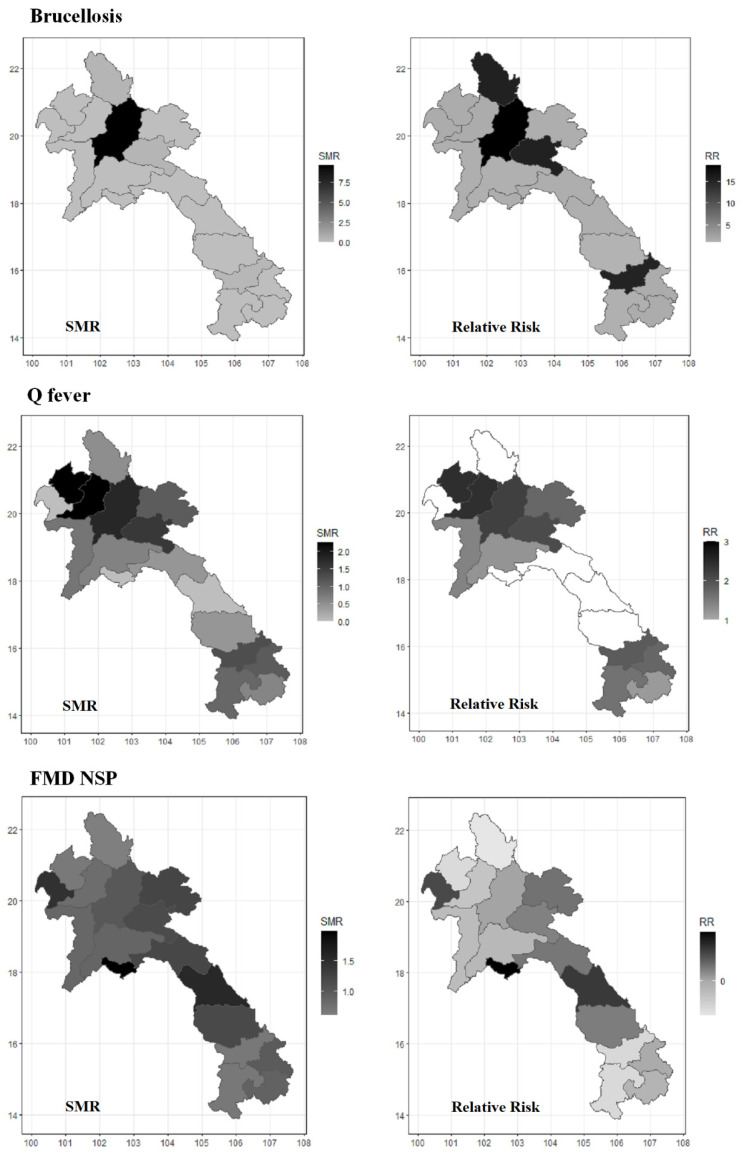
Spatial analysis maps of SMR (**left**); mean posterior relative risk (MPRR, **right**).

**Table 1 tropicalmed-07-00078-t001:** Overall seroprevalence.

Tested Animals (*n* = 4247)	Positive	% Apparent Seroprevalence (95% CI)	% True Seroprevalence (95% CI)
FMD NSP *	2144	50.5 (49.0, 52.0)	54.6 (53.0, 56.2)
Brucellosis	30	0.7 (0.5, 1.0)	0.4 (0.2, 0.7)
Q fever	71	1.7 (1.3, 2.1)	1.7 (1.3, 2.1)

* Foot -and-Mouth Disease Non-Structural Protein.

**Table 2 tropicalmed-07-00078-t002:** Detail of brucellosis seropositive samples.

Province of Origin	Date Collected	Type	Positive
Salavan	28 May 2020	Cattle	1
Xiangkhouang	24 August 2020	Cattle	1
Phongsali	6 September 2020	Buffalo	1
Luang Prabang	6 September 2020	Cattle	3
	7 September 2020	Cattle	8
	8 September 2020	Cattle	3
	9 September 2020	Cattle	4
	26 September 2020	Cattle	3
	27 September 2020	Cattle	3
	28 September 2020	Cattle	3
Total			30

**Table 3 tropicalmed-07-00078-t003:** Seroprevalence per province of origin.

Province of Origin	Type	Total (*n*)	FMD NSP *		Brucellosis		Q Fever	
			Positive	Sero Prevalence	Positive	Sero Prevalence	Positive	Sero Prevalence
Attapeu (ATP)	Buffalo	57	30	52.6%	0	0.0%	1	1.8%
	Cattle	39	14	35.9%	0	0.0%	0	0.0%
Bokeo (BK)	Buffalo	74	53	71.6%	0	0.0%	0	0.0%
	Cattle	36	28	77.8%	0	0.0%	0	0.0%
Bolikamxai (BLX)	Buffalo	106	69	65.1%	0	0.0%	0	0.0%
	Cattle	170	99	58.2%	0	0.0%	2	1.2%
Champasak (CPS)	Buffalo	121	49	40.5%	0	0.0%	0	0.0%
	Cattle	308	98	31.8%	0	0.0%	7	2.3%
Houaphanh (HP)	Buffalo	66	40	60.6%	0	0.0%	0	0.0%
	Cattle	256	160	62.5%	0	0.0%	6	2.3%
Khammouane (KM)	Buffalo	114	85	74.6%	0	0.0%	0	0.0%
	Cattle	181	148	81.8%	0	0.0%	0	0.0%
Luang Namtha (LNT)	Buffalo	11	3	27.3%	0	0.0%	0	0.0%
	Cattle	42	15	35.7%	0	0.0%	2	4.8%
Luang Prabang (LBP)	Buffalo	117	51	43.6%	0	0.0%	1	0.9%
	Cattle	282	154	54.6%	27	9.6%	11	3.9%
Oudomxay (OUD)	Buffalo	114	38	33.3%	0	0.0%	6	5.3%
	Cattle	178	80	44.9%	0	0.0%	5	2.8%
Phongsali (PSL)	Buffalo	181	57	31.5%	1	0.6%	0	0.0%
	Cattle	159	48	30.2%	0	0.0%	3	1.9%
Salavan (SLV)	Buffalo	94	39	41.5%	0	0.0%	2	2.1%
	Cattle	326	106	32.5%	1	0.3%	7	2.1%
Savannakhet (SVK)	Buffalo	110	42	38.2%	0	0.0%	0	0.0%
	Cattle	437	284	65.0%	0	0.0%	4	0.9%
Vientiane prefecture (VTE)	Cattle	1	1	100.0%	0	0.0%	0	0.0%
Vientiane (VTP)	Buffalo	13	5	38.5%	0	0.0%	0	0.0%
	Cattle	53	25	47.2%	0	0.0%	0	0.0%
Xekong (XEK)	Buffalo	44	26	59.1%	0	0.0%	0	0.0%
	Cattle	115	53	46.1%	0	0.0%	3	2.6%
Xiangkhouang (XKH)	Buffalo	46	33	71.7%	0	0.0%	0	0.0%
	Cattle	297	168	56.6%	1	0.3%	9	3.0%
Xaisomboun (XSB)	Buffalo	4	2	50.0%	0	0.0%	0	0.0%
	Cattle	20	8	40.0%	0	0.0%	1	5.0%
Xayabuli (XY)	Buffalo	43	19	44.2%	0	0.0%	0	0.0%
	Cattle	31	13	41.9%	0	0.0%	1	3.2%
Thailand	Buffalo	1	1	100.0%	0	0.0%	0	0.0%
Total		4247	2144	50.5%	30	0.7%	71	1.7%

* Foot-and-Mouth Disease Non-Structural Protein.

**Table 4 tropicalmed-07-00078-t004:** Univariate analyses and *p*-values.

Variable (Reference Group)	*p*-Value ^^,^**	Category **	Odds Ratio (95% CI)
FMD NSP *			
Destination province (Phongsali)	<0.00001	Attapeu	1.8 (1.1, 2.8)
		Bokeo	6.0 (3.7, 9.9)
		Bolikamxai	3.7 (2.7, 5.2)
		Houaphanh	3.6 (2.6, 4.9)
		Khammouane	10.1 (6.9,14.8)
		Luang Namtha	2.2 (1.4, 3.5)
		Luang Prabang	2.5 (1.8, 3.4)
		Oudomxay	1.5 (1.1, 2.1)
		Savannakhet	4.6 (3.4, 6.4)
		Vientiane prefecture	1.9 (1.4, 2.7)
		Xekong	3.0 (2.0, 4.6)
		Xiangkhouang	3.2 (2.4, 4.4)
		Xayabuli	1.9 (1.1, 3.2)
Province of origin (Phongsali)	<0.00001	Attapeu	1.9 (1.2, 3.0)
		Bokeo	6.3 (3.9, 10.1)
		Bolikamxai	3.5 (2.5, 4.9)
		Houaphanh	3.7 (2.7, 5.1)
		Khammouane	8.4 (5.9, 12.1)
		Luang Prabang	2.4 (1.7, 3.2)
		Oudomxay	1.5 (1.1, 2.1)
		Savannakhet	3.3 (2.5, 4.4)
		Vientiane	1.9 (1.1, 3.2)
		Xekong	2.2 (1.5, 3.3)
		Xiangkhouang	3.2 (2.3, 4.3)
		Xayabuli	1.7 (1.0, 2.9)
Sex (Male)	7.97 × 10^−8^	Sex: Female	1.4 (1.3, 1.6)
Age (1–2 Years)	0.009269	Age: 9 Years or more	1.5 (1.1, 1.9)
Date collection (25 January 2020)	<0.00001	Date collected 23 September 2020	10.5 (1.4, 78.1)
		Date collected 6 November 2020	15.3 (1.9, 122.8)
		Date collected 7 November 2020	9.7 (1.4, 65.4)
		Date collected 8 November 2020	11.5 (1.6, 85.2)
		Date collected 20 November 2020	8.0 (1.1, 56.8)
Collection month (March)	<0.00001	Collection.month: August	1.4 (1.0, 1.9)
		Collection.month: September	2.3 (1.7, 3.2)
		Collection.month: October	1.9 (1.3, 2.5)
		Collection.month: November	2.3 (1.7, 3.2)
		Collection.month: December	1.4 (1.1, 2.0)
**Q fever**			
Destination province (Bokeo)	0.005182	-	-
Province of origin (Bokeo)	0.005772	-	-
Animal type (Buffalo)	0.000855	Animal type: Cattle	2.8 (1.4, 5.4)
Collection month (February)	3.40 × 10^−6^	-	-
**Brucellosis**			
Destination province (Bokeo)	<0.00001	-	-
Province of origin (Bokeo)	<0.00001	-	-
Body condition score (score 1)	0.003206	-	-
Age (1–2 Years)	0.007222	-	-
Animal type (Buffalo)	9.87 × 10^−5^	Animal type: Cattle	13.1 (1.8, 96.5)
Collection month (February)	<0.00001	-	-

* Foot and-Mouth Disease Non-Structural Protein; ** only included variables/categories with *p*-value < 0.05, ^ Chi-square.

## Data Availability

Not applicable.

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
