# Peer review of "Abattoir-Based Serological Surveillance and Spatial Risk Analysis of Foot-and-Mouth Disease, Brucellosis, and Q Fever in Lao PDR Large Ruminants"

_tropicalmed, 2022, doi:10.3390/tropicalmed7050078_

Round 1

Reviewer 1 Report

The approach planned and the data generated does not support the seroprevalence claimed and the prediction of the relative risk may not be true

Author Response

With reference to the manuscript the authors claim report the Seroprevalence of Qfever, brucellosis and FMD in large ruminants in a restricted province based on the serum samples collected from abattoir in each of the province

However, the following are the queries that the authors need to clarify to justify their claim or provide significance to the finding:

  • The authors have clearly titled the study as an Abattoir based serological surveillance but however report the results as Seroprevalence for the diseases screened (Q-fever, FMD and Brucellosis). The data generated is only the sero-positivity of the samples collected at abattoirs as it is selected samples screened based on the availability.

Response: Thank you to the reviewer for this comment.We applied the passive surveillance technique where samples were collected based on availibility regularly. The prevalences calculated in our study only represented the abattoir populations presented at slaughter rather than the wider provincial animal population

  • One of the major deficiency in the study – is the number of samples collected over the period of study (January to May had once a month sampling while the remaining period had twice a month). The results of the data presented reveals higher seropositivity in the period sampled twice a month and the lower sero-positivity when sampled once a month (Table 1, Figure 2 & 3; specifically with reference to FMD)

Response: The first five month program was a trial period as field provincial teams had minimum capacity to collect samples from large ruminants. Season may have some affects on the prevalence of the diseases due to agriculture activities (wet and dry). Other factors would be animal trade movements driven by the demand and supply. These two points have been addressed in the discussion section.

  • The authors report the Seroprevalence per province of origin based on the representative sample from the abattoir and it is not a properly planned sampling protocol for the province. Hence the claim of Seroprevalence may not be appropriate(e.g. Vientiane prefecture- the sample available/ positivity for FMD is 1/1 = 100%)

Response: Passive surveillance applied in this study have limitations. The n =1 from Vietiane prefecture (Table3) was the province of origin which is the animal data. The Vietiane prefecture team collected 300+ samples from the abattoir in their province in 2020. However, as Vietiane prefecture is the Lao PDR capital city with high density urban areas, large animals raising in the area is rare.

  • There has been preferential sampling in some areas (Vientiane vs Phongsali – fewer vs higher number of samples in these areas) and the type of animals slaughtered at each site also depended only on the availability and local demand

Response: We agree with the reviewer.Total sample numbers per province were depend on many factors including the number of of animals presented at abattoirs for slaughter and also staff capacity. Both of these factors are largely beyond our control

  • With respect to the Lao PD, the authors indicate that there is no clear data on the vaccination history pertaining to FMD as well as Brucella. The study also has a preferential sampling based on the sample available from the abattoir. With these two major deficiencies, the calculation of the mean posterior relative risk based on the population size may not be perfectly true

Response: Brucella vaccination was not used in Laos. FMD vaccines are available but farmers would have to pay. Thus vaccination rate is relatively low. The relative risk of the regression models was not calculated only the significant ORs were present.

  • With respect to Brucella antibody screening

✓ There is no gold standard method to screen for antibodies to brucella and tests like Rose Bengal Plate Test (RBPT) and enzyme linked immunosorbent assay (ELISA) are not always sensitive or specific due to cross-reactivity with other bacterial antigens.

✓ Hence, most of the commercial kits that are available have been calibrated to define the cut off for negative, weak positive and strong positive as per Council Directive 64/432/EEC.

✓ The study has used RBPT to confirm the ELISA positive brucella antibody samples and some samples have been screened only by RBPT

Response: Thank you for this observation. We agree with the reviewer. RBT and ELISA were used as they are locally available and easy to set up in low capacity setting. Despite two diagnostic tests were used, all positive samples were positive to RBT. We have addressed this issue in the responses to reviewer 1.

Thus there is a clear need of additional data to be generated to claim the Seroprevalence as the relative risk. The data reported is restricted a particular province and will be interesting to the local readers. Hence the article in my view is not suitable for a research publication; it can be included as a letters to editor including all the deficiencies.

Response: Thank you for your opinion. To set up this type of surveillance in a resource poor setting to cover numerous provinces is a large logistical task. We believe we have addressed the limitations of the data generated in the study. For the reviewer to state that the article is not suitable as a research publication is unrealistic. The need to understand disease prevalence in a resource poor setting is very important and the approaches used must be sustainable. It should be noted that relative risks in the spatial modelling were based on the count number of seropositive in the destination (abattoir) province (not the province of origin figures in Table 3). Also Xaisomboun province with minimum number of sample was omitted from the spatial models.

Author Response

Reviewer 2

The "Introduction" part is not clear and does not introduce the aim of the article itself and does not attract the attention of the readers. It is advisable to rewrite the introductory part and to provide the information "about the general topic of the article in the light of the current literature which paves the way for the disclosure of the objective of the manuscript".

Response: The introduction section has been rewritten.

Lines 97-100 - please explain, if the surveillance activities were carried out in the entire country or in 18 provinces, while in the text this part is not entirely clear.

Response: The sentence has been rewritten to "The surveillance was conducted between January and December 2020 in all eighteen provinces (one prefecture and 17 provinces) throughout Lao PDR by provincial teams comprising of trained District Agriculture and Forestry Office (DAFO) and Provincial Agriculture and Forestry Office (PAFO) personnel."

Lines 100-101 - what was the reason to collect blood samples once or twice a month? Does it somehow connect to any of the risk factors? If yes, more explanation is needed.

Response: The sentence has been rewritten to "Blood samples were collected from an abattoir in each province once a month between January and May as a trial period and twice a month from June to December to maximise information."

Lines 112-115 - why thus particular diseases were selected for the testing - Brucellosis, Q fever, and FMD? What was the reason behind it? Why do the authors include only these pathogens in the study? A more detailed explanation would describe more in detail the aim of the study conducted.

Response: the introduction section has been revised as suggested.

In the results section, purely statistical data is presented. From the presented data and discussion remain unclear, why different age groups were found to be seropositive and why the others were not, why seropositive animals were found during the particular month of sample collection, how the seasonality might be related with the disease spread, how and why Q fever or Bruccelosis was related with the age, and why, maybe FMD NSP was linked with the previously vaccinated animals or animals from the vaccinated herds.

Response: Paragraphs were added to the discussion section to Lines 389-400 and 416-419 to address the above points.

Reviewer 3 Report

The "Introduction" part is not clear and does not introduce the aim of the article itself and does not attract the attention of the readers. It is advisable to rewrite the introductory part and to provide the information "about the general topic of the article in the light of the current literature which paves the way for the disclosure of the objective of the manuscript".

Lines 97-100 - please explain, if the surveillance activities were carried out in the entire country or in 18 provinces, while in the text this part is not entirely clear. 

Lines 100-101 - what was the reason to collect blood samples once or twice a month? Does it somehow connect to any of the risk factors? If yes, more explanation is needed.

Lines 112-115 - why thus particular diseases were selected for the testing - Brucellosis, Q fever, and FMD? What was the reason behind it? Why do the authors include only these pathogens in the study? A more detailed explanation would describe more in detail the aim of the study concducted.

In the results section, purely statistical data is presented. From the presented data and discussion remain unclear, why different age groups were found to be seropositive and why the others were not, why seropositive animals were found during the particular month of sample collection, how the seasonality might be related with the disease spread, how and why Q fever or Bruccelosis was related with the age, and why, maybe FMD NSP was linked with the previously vaccinated animals or animals from the vaccinated herds. 

Author Response

(The authors gave the same response as above.)

Reviewer 4 Report

The manuscript is well written and provided intersting data and insights. Besides some minor mistakes, i.e. spelling and typos errors for which the manuscript need to be further carefully checked, the outcome of the figures should be improved. Especially figure 4 should be adapted to provide a better overview about what the authors would like to show. Some figure can easily be adapted by increasing the figures, for other figures the authors should use colors to indicate the interesting things...

Author Response

Reviewer3

The manuscript is well written and provided intersting data and insights. Besides some minor mistakes, i.e. spelling and typos errors for which the manuscript need to be further carefully checked, the outcome of the figures should be improved.

Response: amended as suggested.

Especially figure 4 should be adapted to provide a better overview about what the authors would like to show. Some figure can easily be adapted by increasing the figures, for other figures the authors should use colors to indicate the interesting things...

Response: amended as suggested.

Round 2

Reviewer 2 Report

The author did not reply to any of my comments. I believe that some issue occurred because the authors provide answer but to other comments, not mine’s.

Author Response

Dear Reviewer,

My apologies, I mismatched the responses to reviewers during the resubmission due to the reviewer order change which I had notified the editor immediately. Somehow you did not receive the right response. Please find our response to your comments in the attached file. 

Reviewer 3 Report

In the Result section, the first sentence- check the spelling.

Author Response

Dear Reviewer,

Thank you very much for your comment. I checked through the whole document again for typos. 

Round 3

Reviewer 2 Report

The authors responded to all my comments and made substantial corrections to the text. I have only minor comments before accepting this manuscript for publication. 
Regarding my previous ELISA results comment that the authors did not understand, I was referring to other methods to analyze ELISA results such as ROC, since ELISA provides semiquantitative data. But I understand that you used ELISA as a qualitative result (positive or negative) for the initial screening. You don´t need to reply to this and the way you used the data is ok. 

The new paragraph in the introduction is confusing:

Neglected zoonoses, including brucellosis and Q fever (caused by Coxiella burnetii) have been previously reported in livestock in Lao PDR 35 [6]. However, information on the disease circulations in Lao PDR's livestock populations and potential impacts on humans are still widely 37 unknown. Brucellosis causes productivity losses, including abortion, 38 infertility, milk reduction and unhealthy calves [7]. In ruminants, Q fever 39 often causes mild disease, but in some cases, abortion and stillbirths can  occur [8]. Brucellosis can be transmitted to humans by direct contact via 41 mucous membranes or open wounds and ingestions of infected products 42 [9], while C. burnetii transmitted by inhalation of contaminated dust or 43 drinking unpasteurised milk

Please, mention one disease and then talk about the other: 
Neglected zoonoses, including brucellosis and Q fever (caused by Coxiella burnetii) have been previously reported in livestock in Lao PDR 35 [6]. However, information on the disease circulations in Lao PDR's livestock populations and potential impacts on humans are still widely unknown. In ruminants, Q fever often causes mild disease, but in some cases, abortion and stillbirths can occur [8]. C. burnetii can transmitted to humans by inhalation of contaminated dust or drinking unpasteurised milk. Brucellosis causes productivity losses, including abortion, infertility, milk reduction and unhealthy calves [7], and can be transmitted to humans by direct contact via mucous membranes or open wounds and ingestions of infected products [9], 
Here, specific buffalo papers should be cited. 
In addition, I miss some important references about the specific diseases in Buffalo. I see some FAO references that only deals with the general aspect of the disease and some references if another species. Recent papers dealing specifically with the buffalo specie should be cited, such as:

Q fever:
https://doi.org/10.1371/journal.pone.0239260
https://doi.org/10.1371/journal.pone.0192188

Brucellosis:
https://doi.org/10.1111/tbed.13192

FMD:
https://doi.org/10.1016/j.prevetmed.2021.105318

Table 4 has some problems. 
Do not use this number format: < 2.2x10-16
p-value** (Chi-square)
This may confuse readers. Just inform P-value and explain that it refers to Chi-square in the table footnote. 
Since the P-value is very very low, just use: < 0.00001 

2.5% and 97.5% are related to what. Hard to understand this. Probably because of the track changes. 

The information about the odds ratio must be moved to the table footnote. 

Author Response

Thank you very much for your valuable comments, please find the response specific to each comment in the attached file.
